# LRFMV: An efficient customer segmentation model for superstores

**Rezwana Mahfuza**[1], **Nafisa Islam**[1], **Md. Toyeb**[1], **Md Asaduzzaman Faisal Emon**[1], **Shahnur Azad Chowdhury**[2‡], **Md. Golam Rabiul Alam**[1‡]*

**1** Dept. of Computer Science and Engineering, Brac University, Dhaka, Bangladesh, **2** Dept. of Business Administration, International Islamic University Chittagong, Sonaichhari, Bangladesh

☯ These authors contributed equally to this work.
‡ SAC and MGRA also contributed equally to this work.
* rabiul.alam@bracu.ac.bd

**Data Availability Statement:** This research used the global superstore dataset of Tableau community that is a publicly available open access

## Abstract

The Recency, Frequency, and Monetary model, also known as the RFM model, is a popular and widely used business model for determining beneficial client segments and analyzing profit. It is also recommended and frequently used in superstores to identify customer segments and increase profit margins. Later, the Length, Recency, Frequency, and Monetary model, also known as the LRFM model, was introduced as an improved version of the RFM model to identify more relevant and exact consumer groups for profit maximization. Superstores have a varying number of different products. In RFM and LRFM models, the relationship between profit and purchased quantity has never been investigated. Therefore, this paper proposed an efficient customer segmentation model, namely LRFMV (Length, Recency, Frequency, Monetary and Volume) and studied the profit-quantity relationship. A new dimension V (volume) has been added to the existing LRFM model to show a direct profit-quantity relationship in customer segmentation. The V stands for volume, which is derived by calculating the average number of products purchased by a frequent superstore client in a single day. The data obtained from feature extraction of the LRMFV model is then clustered by using conventional K-means, K-Medoids, and Mini Batch K-means methods. The results obtained from the three algorithms are compared, and the K-means algorithm is chosen for the superstore dataset of the proposed LRFMV model. All clusters created using these three algorithms are evaluated in the LRFMV model, and a close relationship between profit and volume is observed. A clear profit-quantity relationship of items has yet not been seen in any prior study on the RFM and LRFM models. Grouping customers aiming at profit maximization existed previously, but there was no clear and direct depiction of profit and quantity of sold items. This study applied unsupervised machine learning to investigate the patterns, trends, and correlations between volume and profit. The traits of all the clusters are analyzed by the Customer-Classification Matrix. The LRFMV values, larger or less than the overall average for each cluster, are identified as their traits. The performance of the proposed LRFMV model is compared with the legacy RFM and LRFM customer segmentation models. The outcome shows that the LRFMV model creates precise customer segments with the same number of customers while maintaining a greater profit.

dataset (available at: https://data.world/asepetruk/global-superstore).

**Funding:** The author(s) received no specific funding for this work.

**Competing interests:** The authors have declared that no competing interests exist.

## Author summary

### Why was this study done?

- Superstore business has been booming in the last decades. In the FY-2017, retail revenue of the top 250 superstores was 4,530,059 million USD which achieved 5.7% economic growth [1].

- The Length, Recency, Frequency, and Monetary model, also known as the LRFM model, was introduced as an improved version of the RFM model to identify more relevant and exact consumer groups for profit maximization. However, there exists a substantial association between the purchase quantity and revenue generation that had been overlooked in earlier models. In this research, we introduced the LRFMV model, an improved version of existing business models for superstores, to further assess how much revenue boost and marketing strategy can be developed for the superstore industry and contribute to both technical sectors and the business world.

- In this research, we searched for a new way to utilize the segmentation model based on the scoring procedure and encountered how a business based matrix can employ them to have a substantial influence on the existing collaborative business and technology sector.

### What did the researchers do and find?

- We proposed an efficient customer segmentation model, LRFMV and tried to observe the profit-quantity relationship. A new dimension $V$ (volume) has been added to the existing LRFM model in order to show a direct profit-quantity relationship. Here, the $V$ stands for volume, which was derived by calculating the average number of products purchased by a frequent superstore client in a single day.

- To get the final average as volume in a specified time frame, the previously found average amount was divided by total days in a limited period of time of visitation of that customer. Quantity of purchased goods refers to the average amount of procured product by repeatedly going customers.

- Superstores have a varying number of different products with the record of being bought in different quantities multiple times on the same day by a specific customer. In RFM and LRFM models, the relationship between profit and purchased quantity and how they can contribute to an effective customer behavioral analysis was not investigated and evaluated.

### What do these findings mean?

- It is visible that a large volume of purchased products positively influences the profit maximization of a superstore.

- The establishment of the proposed model will assist superstores in generating more profit and performing comprehensive business analysis by helping to find the most profitable group of customers.

# 1 Introduction

## 1.1 Background and motivation

The technique of grouping or segmenting a customer base based on shared qualities such as age, gender, preferences, and shopping patterns is known as customer segmentation. This allows businesses to recognize certain smaller customer groups for targeted marketing, especially to boost the possibility of customers making a purchase. The RFM segmentation, like other segmentation approaches, is useful for identifying customer groups that require specific attention.

The term "superstore" refers to a store that is self-service and physically organized with multiple divisions to ensure a variety of food and grocery goods [2]. This retail type is often smaller than a hypermarket but larger than a standard grocery store.

A superstore's customers are generally local individuals and small enterprises that need home goods replenishment on a recurrent basis. The suppliers to a superstore are typically manufacturers of household items located in regions far from the ultimate customers. In effect, the superstore acts as a virtual marketplace, bringing faraway suppliers and local customers together. The superstore's "product" under this setup is its supply chain [3], and the main essence of the business model is to get as many individuals from all socioeconomic backgrounds through the door as possible and to keep them.

Observers and retail entrepreneurs agree that the superstore network in this fast-growing globe is insufficient due to increased urbanization. As market consumption habits changed and actual incomes climbed, superstore chains grew in phases worldwide, not only in first-world economies. It is critical to keep track of clients' shopping behaviour to expand these stores. In addition, due to the covid-19 pandemic, most business sectors faced severe economic crises; this superstore retail business was no exception. According to Koren et al., the social distancing from February to May 2020 caused by the Covid-19 pandemic reduced the wages of the workers by 39.9 percent [4].

According to Desouza et al., [5], organizations may use segmentation to distinguish and categorize customers depending on those characteristics, allowing them to define target audiences. These characteristics, when handled properly, can enhance both the service and the product.

The study's goal is to overcome this gap by developing a customer profile system tailored to real-world retail customers. It recommends a hybrid solution focusing on clustering for client profiling in superstore operations, in particular. This feature uses the LRFMV model to analyze customer values before profiling them using the partition based clustering approach, an unsupervised learning approach. Unsupervised learning occurs when a computer receives $x = \{x_1, x_2. \ldots \ldots .x_N\}$ but receives no supervised target outputs or incentives from its surroundings.

A partitioning or clustering method begins by creating an initial set of k partitions, where k is the number of partitions to construct. It then employs an iterative relocation strategy, in which objects are moved from one group to another in an attempt to enhance partitioning. The criterion that distinguishes partition-based algorithms is that objects in the same cluster are close or related to each other, whereas objects in other clusters are far apart and unrelated and this makes partition-based algorithms different and more popular from other clustering algorithms. The data points are partitioned into one level with some clustering approaches. K-Means, Mini-batch, and K-Medoid are examples of such techniques.

Despite the presence of a large range of clustering algorithms, k-means, k-medoids, and mini-batch were chosen for this research work after examining some key functionalities that are closely related to the dataset in this concern. K-means algorithm is known for its speedy execution time and scalability. The Mini Batch K-Means algorithm is a reliable and efficient

method for clustering with similar characteristics while lowering the costs of processing such massive amounts of data. In the case of K-medoids, instead of using the centroid of the items in a cluster as a reference point as k-means clustering does, clustering uses the medoid as a reference point. A medoid is an object in the cluster that is most centrally placed or has the least average dissimilarity to all other objects which makes K-medoids more noise-resistant. Moreover, these three algorithms ensure that the same data points do not belong to multiple clusters. Since they do hardly allow repetitive data points in different clusters, these partitioning techniques are relatively redundant in terms of cluster nodes.

In the case of K-means clustering, the algorithm has successfully completed the clustering procedure or grouping of data points in k number of clusters for the given dataset. If the centroids (k values) in k-means remain at the same position or in point for two iterations, the algorithm ensures that it has completely grouped the data points into correct clusters. It can simply be adapted to new data sets and new examples can be generated. It can be applied to clusters of all forms and sizes, including elliptical clusters.

K-Medoids clustering is a partition-based technique that includes the ability to generate empty clusters and has the sensitivity to outliers or noise. It also ensures convergence, identical to the k-means approach as well as determining the cluster's member with the most coherence. Finding a cluster is made easier and cuts computational costs with the Mini-batch K-means technique. The mini-batch strategy was one of the chosen approaches for this paper since the dataset was large enough and will continue to grow in the future for this specific business market.

One of the goals of this study is to compare several algorithms in order to achieve the best possible result, and these three algorithms demonstrated their efficacy for our target dataset and business. We found closely related outputs when we compared the findings of K-Means, Mini-batch, and K-Medoid for our dataset, ensuring clustering accuracy.

An in-depth comparison of RFM analysis, LRFM analysis and LRFMV analysis is included in the study. RFM analysis is a type of quantitative analysis that allows a corporation to categorize its consumers based on their behaviour. Recency, Frequency, and Monetary are abbreviated as RFM. Several studies have found that by adding another component to the basic RFM Model, the predictability of customer behaviour can be increased. According to Wu et al., [6], the LRFM model was built based on the RFM model, a well-known technique for analyzing customer values for market research. Reinartz et al. [7] discussed the RFM model's inability to differentiate between long- and short-term partnerships with customers. Customer satisfaction is determined by the partnership between a business and its customers and is built over time by effective customer relationship management [8]. By simply adding the customer (V) volume to the RFM Model, customer segments become more efficient and precise. In this research, the influence of the proposed LRFMV model on superstores has been analyzed, along with a comparison between the traditional RFM model and the proposed LRFMV model.

## 1.2 Contributions

The proposed methodology demonstrates a novel technique for using unsupervised machine learning to develop a commercial solution. The key contributions of this research are the presentation of the correlation between the volume of products and the profit earned against each customer along with the findings from the evaluation of the LRFMV model after applying the standard K-means, K-Medoids, and Mini Batch K-means algorithms. Moreover, it compares the result of LRFMV analysis with traditional RFM and LRFM analysis. The following summarizes the major contributions of this research.

- This research introduces a new dimension V (volume) to the existing LRFM model in order to show a direct profit-quantity relationship in customer segmentation. The suggested method found a high connection between profits per head and the commodity bought by each client in a single transaction. The LRFMV model is capable of addressing a number of problems connected to identifying the optimum client for the optimal product.

- The proposed method has a unique blend of data preprocessing, scoring system, segmentation, and concluding results from the segments using a matrix. At first, some new features on their purchasing habits were analyzed using existing features where the existing features themselves were not picked for segmenting the customers in this case. The data was then reduced into five main features Length (3.3.1), Recency (3.3.2), Frequency (3.3.3), Monetary (3.3.4), and Volume (3.3.5) with their individual equations, which are identical to the scoring system. After that, the dataset was segmented based on the score.

- A comparison of different types of centroid-based clustering algorithms is also shown based on this model. Some famous clustering algorithms are K-means, Mini-Batch, and K-medoids, which belong to the same type but they work in different mechanisms. K-means minimize total squared error, whereas k-medoids tend to minimize the sum of dissimilarities between points. Mini-batch works like the K-Means algorithm with the fixed size of chunks. The differences in mechanism between these three algorithms also affect the end result of our model, which is clearly mentioned.

- Finally, a customer classification matrix was used in the suggested system to analyze the results with profit margin, revenue, and cost to serve for more insights.

### 1.3 Paper organization

The next section contains a literature review and an overview of the related works. The third section contains a detailed discussion about the proposed model. It consists of data collection, data preprocessing, proposed workflow, feature extraction, and model specification. The fourth section describes the comparison between RFM and LRFM model after applying K-means, K-Medoids and Mini Batch K-means algorithms, along with a discussion on the analysis of the results. It describes the statistical result analysis of each clustering algorithm applied in this study. Finally, the last section is the conclusion. It contains a research overview, contribution, impacts, and future works.

### 2 Literature review

Over the years, many data mining techniques have been widely applied in a variety of fields. As the volume of transactions in the business grows, it has become more difficult to segment profitable customers to enhance sales. "Quantity" has been incorporated in different studies for profit maximization as well as association rule mining [9, 10]. The volume or quantity produced directly relates to the revenue, cost, and profit function [11]. Lau. J. [12] explored several applications of profit function through an alternate derivation to characterize production and profit function. According to the study, the firm maximizes profit with the optimized quantities of the variable inputs where maximization of the profit is similar to the normalized profit function. Chen. Chung-Ho [13] et al. presented the supply chain system's optimal profit-sharing model between producers and consumers. According to the study, the order quantity contributes to the average consumer demand and significantly impacts the predicted profits of the supplier and the buyer.

In the customer segmentation process, the RFM model can be applied to the purchase experience of a customer, the development of improved prediction and classification techniques [14, 15]. To optimize marketing results, both customer segmentation and customer targeting are needed [16]. The transactional data is first subjected to an RFM analysis [17, 18], after which clustering techniques like standard K-means [19–21], Fuzzy C-means [22], and Repetitive Median based K-Means (RM KMeans) clustering algorithms [23] are used. Following that, the clusters are further evaluated to segment customers appropriately.

For better forecasts and identifications, numerous researchers used RFM analysis. For instance, customer satisfaction [24], customer lifetime value [25], churn prediction [14, 26], CLV measurement [27] predict a customer's response to direct marketing [16] can be analyzed using data mining models and customers can be classified according to their profitability. Some authors used the RFM model to create a two-phased model mechanism that can be thought of as a new segmented solution [28]. In addition to superstore research, RFM is also used in a variety of sectors, including banking and insurance [27, 29], telecommunications [30], political score generation [31], on-line industries [32], travel agencies [33], retail industry [34], medical field [14, 35], and so on.

The study conducted by Tavakoli et al. [36] includes the importance of behaving with customers according to their background and category, which has evolved significantly in recent years. The optimal number of clusters is first determined using the self-organizing maps system (SOM) by Daoud et al. [37]. After determining the best number of clusters, the K-means algorithm is used for clustering customer data after performing RFM analysis for each customer of an online selling company in Morocco. Cheng et al. combined the quantitative value of RFM attributes and the K-means algorithm into the rough set theory RS theory to mine classification rules [24]. Using weighted frequent pattern mining, Cho et al. [38] proposed a personalized recommendation scheme in which the RFM model is used to categorize prospective customers. Therefore, many authors have previously regarded the RFM model as the most successful way of better customer understanding and categorization.

To strengthen data mining techniques in customer analysis, Bachtiar [39] proposed two-step mining method based on the RFM model. The obtained data is first analyzed using the RFM technique and later on segmentation using K-means clustering. IF-THEN rules describe customer characteristics after the clusters are further examined using association analysis. Silhouette and Connectivity measures are used to assess the cluster results and evaluate accordingly. Because of using the two steps approach for customer analysis, accurate insight on customer behaviour as well as purchasing tendency are gained, which can help to improve marketing strategy significantly. Only frequent mining patterns have been used in the association study, as irregular patterns are not obtained from the clusters. As a consequence, useful customer research data could be lacking at this stage. Furthermore, enhanced RFM models could strengthen the research process by analyzing profitable customers more precisely as some additional variables are taken into account.

Many researchers started to develop modern RFM models to see how they outperform existing RFM models by including additional variables. For instance, I.-C. et al. proposed the RFMTC model (Recency, Frequency, Monetary, Time after first purchase, Churn probability), which can quantify the likelihood of the buyer repeating the purchase as well as the estimated value of the cumulative amount of possible sales [40]. The RFM model was extended by Soeini et al. [41] by adding two more variables: duration (L) and cost (C). Wei et al. [35] concentrated on the LRFM, which is an expanded RFM that contains a new variable called Length. It refers to the number of days between a client's first and last appointment at the clinic. They also formulated their marketing techniques and created twelve clusters for a total of 2258 dental patients using the SOM (self-organizing maps) methodology. The average LRF values for each

cluster and all patients are estimated, and the values that are higher than the average are analyzed, indicating core patients. The result is better than the traditional RFM model, but there are also some limitations. It is observable that the papers mentioned above have not stated the interconnection between profit and their proposed model as increasing profit is the main goal in most of the business.

In specialized clinics, Mohammadzadeh et al. [14] discovered opportunities for repeat clients to maximize profit and minimize patient failure costs. They used the RFML model to forecast new client turnover and conduct behavioural research on specific existing patients. Following that, they applied the K-means clustering algorithm to group clients and compared different groups of three clinics. A decision tree classifier was also used as a churn predictor based on which the number of faithful and turnover patients were identified. Conversely, the suggested approach may not be effective in all cases due to demographic and regional factors.

Sarvari et al. [42] used a variety of data mining methods, including the K-means algorithm, Apriori association rule mining (ARM), and neural networks, to develop WRFM (Weighted RFM), a revised RFM process. They analyzed the customer data of a global pizza restaurant chain in Turkey by considering demographic data with RFM variables. The cluster is analyzed using both an unweighted conventional RFM value and a weighted WRFM value, demonstrating that adding demographic variables into account yields an excellent outcome with positive associations. Both the Kohonen algorithm (SOM) and the K-means algorithm have been studied and compared for customer segmentation. Between the two clustering algorithms used by the author, K-means produced more consistent results in cluster consistency and runtime. The proposed model shows how WRFM combined with clustering strategies outperforms conventional RFM and strengthens marketing strategy, resulting in increased profits for the business. More demographics characteristics should be considered and compared to standard RFM to see how they work. Aside from that, authors could consider additional variables to systematically observe customers' purchasing patterns.

A combined approach of clustering and association rule mining is proposed by Guney et al. [43]. For the segmentation process, the authors used modified RFM, LRFMP, alongside K-means and apriori algorithms to generate association rules. With the traditional RFM model, L (Length) and P (Periodicity) are introduced by which customer value can be measured successfully. This proposed LRFMP model aims to help define the characteristics and regularity of customers with various purchasing patterns in the grocery and retail sectors [34]. The authors [43] used this model to obtain the values of subscribers using the K-Means algorithm and determined suitable subscribers based on actual customer data from IPTV service providers. Subscribers are divided into four groups after the report determines the most appropriate cluster number with the help of the K-means algorithm: "high consuming-valuable subscribers", "less consuming subscribers", "less consuming-loyal subscribers", and "disloyal subscribers". Although using modified RFM in their research purposes identified potential subscribers, VoD transaction records are obtained via only STB devices. As a result, transactional data using other devices are not considered here. Besides, the approach is limited to Turkish customers, for which it can not be said that the approach will be applicable in all geographic areas where the model should be more generalized and flexible.

## 3 Proposed customer segmentation model (LRFMV)

### 3.1 Proposed system model

Finding a reliable and appropriate data set for the intended outcome was the first step in this project. Then, through a series of stages, this dataset has been preprocessed. Tableau Software

has published the dataset online, which has been used. The dataset was not preprocessed because it had many null and arbitrary values that could stymie the research.

Fig 1 describes the top-level layout of the proposed LRFMV model. Preprocessing consists of a few steps known as Data cleaning, data dimensionality reduction & feature selection and data transformation. After preprocessing, feature extraction was done of that dataset by calculating the desired L, R, F, M, V components for establishing the LRFMV model. The Data frame joining process has been materialized to combine L, R, F, M, V together. The feature reduction step was for reducing the 6 cluster representation to 2 dimensions to avoid complexity. Cumulative Explained Variance and PCA have been used for feature reduction and

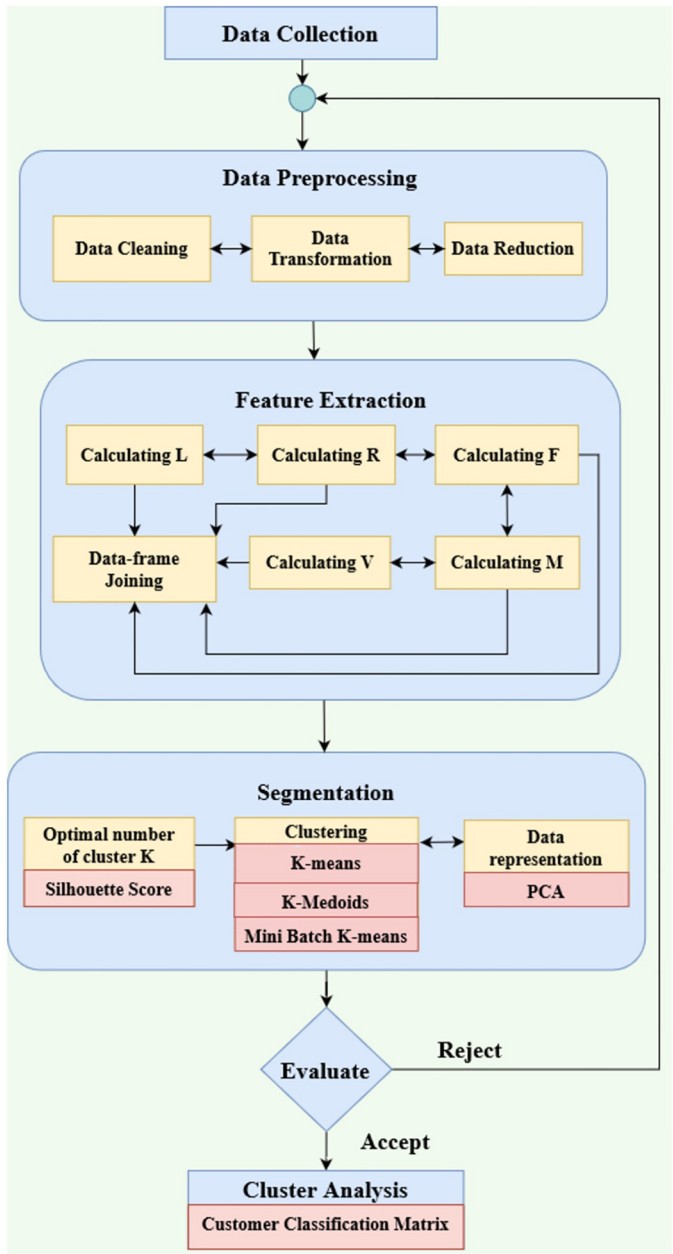

**Fig 1. Top level layout of the proposed LRFMV model.**

representing the dataset in two dimensions. The clusters of customers have been generated using the K-means, K-medoids & Mini Batch K-means algorithms and evaluated by comparing those with the LRFM and RFM model. In this way, evaluation took place. Six efficient clusters for the dataset from those previously mentioned algorithms have been demonstrated. Afterwards, those clusters have been apprised by profit-volume correlation, calculating profit for different clusters, calculating the number of customers in each cluster etc. The optimal and efficient number of clusters has been found for the K-means algorithm using the Silhouette method.

Customer segmentation is assessed using a customer classification matrix and labelled the customers as aggressive, passive, bargain-basement and carriage trade. For failing in cluster evaluation, it was started from the data preprocessing step again.

## 3.2 Dataset description and data preprocessing

Superstore business has been a growing and profitable business for the last few decades worldwide, and the process of finding potential and profitable customers will be beneficial for this sector. This research aimed to scrutinize the cluster of potential buyers by preprocessing and analyzing a standard superstore dataset. This research used the global superstore dataset [44] of Tableau community that is a publicly available open dataset. The Global Superstore data set has around 50000 records, and it is a customer-driven informational dataset that contains comparative data of orders placed through various sellers and markets from 2011 to 2014. Customer order shipment date, city, category, and order priority are among the 24 attributes included in the dataset. The presented customer names and addresses in the dataset are fictional.

A bar chart has been plotted in order to have a glimpse of that superstore data set before preprocessing. The mentioned bar chart in Fig 2 has been plotted with respect to date and various subcategories. Sales of different types of products on different dates are clearly noticeable to understand the earned revenue, and it helps to analyze profit margin.

Missing values are replaced, and inconsistent data is corrected using data cleaning techniques. Each of these activities is carried out in various ways, each of which is tailored to the user's preferences or problem set. For data cleaning, the central tendency for attributes to replace the missing value has been used [45]. In the dataset, many arbitrary and null values created a barrier in establishing the proposed model. So, the mean or average value has been found for those columns where null values occurred frequently. The main aim of this research paper is to add a new component volume to the existing LRFM model. Many attributes need to be dropped from the raw dataset to reach the goal, and unsupervised learning allows us to drop the inconsequential attributes and select the important features.

## 3.3 Feature extraction

The LRFMV model (length, recency, frequency, monetary, and volume) is a simple but powerful tool for market segmentation. LRFMV analysis will segment the customer base and maximize the purchase response rates of marketing efforts, according to this paper. LRFMV research enhances market segmentation by looking at how long (length), when (recency), how much (frequency), how much money (monetary), and how much money a customer spends (volume). According to the study, customers who had recently invested a lot of money and purchased a lot of items were much more likely to react to potential promotions. As a result, the scope of LRFMV research has been broad [19]. Through length, recency, frequency, monetary and volume, the customer relationship matrix assists management in identifying the characteristics of four different types of customer traits [39]. The volume highlights the customers

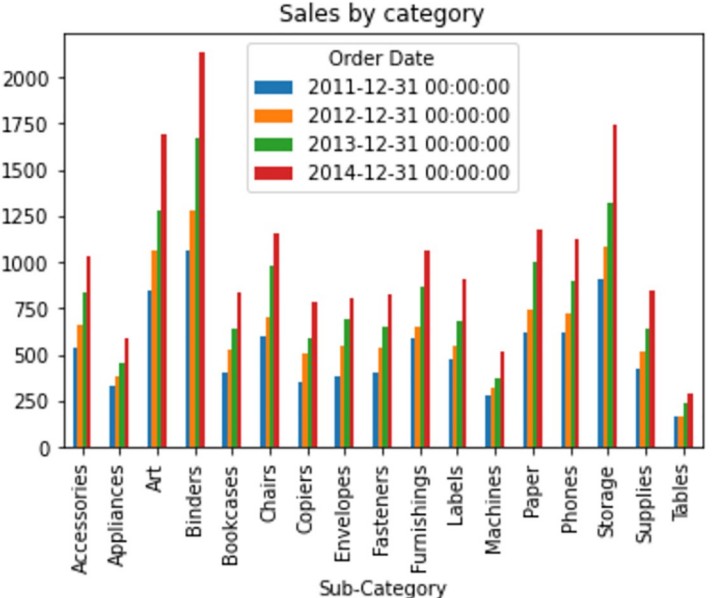

**Fig 2. Barchart of sales by category based on order-dates.**

who provide more profit to the organization as their buying habit is larger than any other customer segment.

**3.3.1 Computation of L.** The number of days between a customer's first and last visit is referred to as length in the LRFMV model. It denotes the distance between two specific visits, or more specifically, it refers to the purchases made on these two dates. Mathematically, if the last purchase date and the first purchase date for a particular customer are denoted by $p_l$ and $p_f$ respectively, then Length, $L$ can be calculated as,

$$L = p_l - p_f \tag{1}$$

In Table 1, it can be observed that Length (L) has been calculated with respect to a customer id. After the customer has completed their first and last purchase, the length of these two purchases has been calculated. It is, in essence, the time interval between two consecutive sales. So,

**Table 1. Determined Length (L).**

|  | Customer ID | Length |
|---|---|---|
| **0** | AA-10315 | 1363.0 |
| **1** | AA-10375 | 1344.0 |
| **2** | AA-10480 | 1212.0 |
| **3** | AA-10645 | 1407.0 |
| **4** | AA-315 | 1300.0 |
| . . . | . . . | . . . |
| **1585** | YS-21880 | 1300.0 |
| **1586** | ZC-11910 | 0.0 |
| **1587** | ZC-21910 | 1192.0 |
| **1588** | ZD-11925 | 1117.0 |
| **1589** | ZD-21925 | 1408.0 |

if all the individual's purchases have been added between these two times, the Length (L) can be found.

**3.3.2 Computation of R.** Recency refers to the days after the last visit of any particular customer. It indicates the days which exist after any valuable customer's last purchase to find the irregularity following that visit. The Recency (R) value was calculated. Mathematically, If the most recent date of the dataset is denoted by $D_r$ and the last purchase date of a particular customer is $C_r$ then Recency, $R$ can be calculated as,

$$R = D_r - C_r \tag{2}$$

In Table 2, the above Eq (2) has been used for calculating the Recency (R) for each customer id. To do so, firstly, the most recent date in the dataset had been identified and saved as a variable, then only the most recent dates of each customer had been stored in a data frame. Then each customer's most recent visit/purchase date was subtracted from the dataset's most recent date, and Recency was assigned to this data.

**3.3.3 Computation of F.** The number of purchases made by a customer in a customer life cycle is referred to as frequency. Counting the number of times a customer purchased any service from the superstore yielded the Frequency (F) value. Mathematically, If the purchase for a customer is denoted by $p_f$ then Frequency, $F$ for that particular customer will be,

$$F = count(p_f) \tag{3}$$

Here, by the count method, the total number of purchases for a particular customer is calculated. The frequency per customer id is being calculated by using this formula in the above-stated Table 3.

**3.3.4 Computation of M.** The total number of transactions and total expenditure were essential to calculate the monetary. In a customer's life cycle, the total amount of his complete transactions has been divided by the number of those transactions to find the monetary value. Mathematically, if the total spending on purchasing of a customer is $p_s$, x is the total number of transactions denoted here. Monetary, $M$ can be calculated as,

$$M = \frac{\sum_{n=1}^{x} p_s}{x} \tag{4}$$

By using the above Eq (4), the monetary value for each customer id is calculated with the dataset for the proposed system model. As a result, from Table 4, the mean value per sale was decided to be used as the monetary value (M) for the LRFMV model.

Table 2. Determined Recency (R).

|  | Customer ID | Recency |
|---|---|---|
| **14418** | AA-10315 | 8 |
| **13931** | AA-10375 | 6 |
| **10701** | AA-10480 | 125 |
| **10970** | AA-10645 | 28 |
| **22935** | AA-315 | 2 |
| . . . | . . . | . . . |
| **4056** | YS-21880 | 9 |
| **21391** | ZC-11910 | 200 |
| **9147** | ZC-21910 | 3 |
| **5217** | ZD-11925 | 3 |
| **9134** | ZD-21925 | 1 |

**Table 3. Determined Frequency (F).**

|  | Customer ID | Frequency |
|---|---|---|
| **738** | AA-10315 | 19 |
| **572** | AA-10375 | 23 |
| **702** | AA-10480 | 20 |
| **21** | AA-10645 | 36 |
| **1113** | AA-315 | 7 |
| . . . | . . . | . . . |
| **410** | YS-21880 | 26 |
| **1589** | ZC-11910 | 1 |
| **13** | ZC-21910 | 37 |
| **902** | ZD-11925 | 9 |
| **708** | ZD-21925 | 20 |

**3.3.5 Computation of V.** In today's world, it has become crucial to combine technical innovation and knowledge with business in order to create and explore new business opportunities. With this in mind, this paper attempted to establish a new technological component that would allow a company to enhance its profitability. With the existing LRFM model, a new feature V has been added with the goal of identifying the most profitable cluster of consumers that will ensure and offer a higher profit to a business. It can be used in a variety of industries to locate potential clients by analyzing their purchasing habits or tendencies.

Volume is a rescaled version of the number of goods purchased by a potential customer. It identifies a group of valuable customers for any company by highlighting the amount of their purchased product over a set period of time or a set number of visits. The proposed term by us in this research paper adds value to the LRFM model by figuring out the customer clusters that give more profit to an organization. It shows that if a customer buys a large amount of product in his certain visits regardless of the spent money, he will contribute more to the profit of that organization.

The higher the number of useful features in a business model, the higher the amount of information can be extracted from a dataset. Hence, we can have more freedom and flexibility in terms of choosing features for clustering. However, we must be cautious when selecting characteristics since if the features are irrelevant, our outcome will be ambiguous, incorrect, and noisy, potentially affecting decision-making. If the feature is relevant, on the other hand, it

**Table 4. Determined Monetary (M).**

|  | Customer ID | Monetary |
|---|---|---|
| **0** | AA-10315 | 732.786474 |
| **1** | AA-10375 | 256.694391 |
| **2** | AA-10480 | 1010.825989 |
| **3** | AA-10645 | 487.490519 |
| **4** | AA-315 | 328.076571 |
| . . . | . . . | . . . |
| **1585** | YS-21880 | 807.958308 |
| **1586** | ZC-11910 | 7.173000 |
| **1587** | ZC-21910 | 789.604683 |
| **1588** | ZD-11925 | 327.914000 |
| **1589** | ZD-21925 | 529.015645 |

allows us to assess and make decisions based on more dimensions that were previously disregarded. Volume is a distinct feature that does not overlap with established business model elements. By adding "Volume" with the existing business models, not only the segments with potential customers have been recognized but also the negative profit driving segments have been identified which can be seen in Fig 13. Furthermore, the segment numbers for the new LRFMV model are higher than for previous models, allowing us to clearly identify distinct sorts of consumer segments with successful business solutions which is visible in Table 7.

From the dataset, the features which solely focus on profit maximization have been considered. The higher the number of purchased products by the customer, the higher the chance of maximizing profit. Therefore, there is a direct relationship with the profit maximization and purchase quantity in any business corporations. We normalized the number of products purchased by a potential consumer over a certain length of time or a set number of visits to determine a group of valuable customers for superstores. When popular existing RFM and LRFM models are compared to the LRFMV model in section 4.1, more categories and new enterprise insights can be generated, and the LRFMV model beats the prior ones. Three clustering strategies were applied for improved clarity, and the segments of LRFMV were able to produce more significant results in each technique. Following that, it is revealed that the higher the increase in volumes, the higher the profit for the majority of the segments which can be seen in section 4.2. Therefore, there exists a direct relationship between profit and volume in the case of superstores which can be a potential key to implement some effective business solutions. As a result an appropriate customer segmentation can be achieved using different clustering algorithms and further analyzed using classification matrices.

Mathematically, suppose the quantity of purchased products for a particular customer is Q. In that case, x is the number of transactions made by a specific customer on a specific day while calculating the average of different attributes, n is the number of days while transacting in his customer life cycle. Therefore, the equation of $V$ can be written as,

$$V = \frac{\sum_{i=1}^{n} \left( \frac{\sum_{j=1}^{x} Q_j}{x} \right)_i}{n} \tag{5}$$

This paper calculates the mean quantity by grouping the dataset with both the customer id and the sale date. It will provide us with the average quantity of products purchased by each customer on specific days. The quantity's mean will be calculated once more. This time, however, it will be grouped solely by customer id. It will be referred to as Volume (V). In Table 5, the volume had been calculated for each customer id using the newly proposed volume formula.

Table 5. Determined Volume (V).

|  | Customer ID | Volume |
|---|---|---|
| 0 | AA-10315 | 3.429825 |
| 1 | AA-10375 | 3.470290 |
| 2 | AA-10480 | 4.060000 |
| 3 | AA-10645 | 3.783333 |
| 4 | AA-315 | 2.285714 |
| . . . | . . . | . . . |
| 1585 | YS-21880 | 3.992222 |
| 1586 | ZC-11910 | 1.000000 |
| 1587 | ZC-21910 | 4.012831 |
| 1588 | ZD-11925 | 2.922222 |
| 1589 | ZD-21925 | 3.287500 |

Transforming the data into a format suitable for Data Modeling is the final phase of data preprocessing. This step is conducted to change the data into a format used in the mining process. In the data transformation phase, a set of attributes (Length, Recency, Frequency, Monetary, Volume, K-means, K-Medoids, Mini Batch K-means etc.) is used to create new attributes. Additional attributes are built from the supplied collection of attributes in this technique to aid the mining process.

To create new attributes, summary and aggregation operations are conducted to a set of attributes. The extracted L, R, F, M, V has been rescaled with the data standardization technique in such a way that the mean value of the attribute is 0 and the standard deviation for the resultant distribution is 1.

## 3.4 Model specification

For the model specification for this study, some prominent approaches such as the Silhouette coefficient, Cumulative Explained Variance Ratio, and PCA were employed. Each method contributed to the identification of certain independent variables that required to be introduced in the research.

**3.4.1 Silhouette coefficient.** The silhouette coefficient is a statistic used to determine the quality of a clustering process. Its value is between -1 and 1.

1: Indicates that clusters are firmly separated and distinct from one another.

0: Indicates that clusters are indifferent or that the distance between clusters is negligible.

-1: Indicates that clusters have been allocated incorrectly. Mathematically, silhouette coefficient can be written as,

$$S(i) = \frac{c(i) - d(i)}{\max\{c(i), d(i)\}}, if |X_i| > 1 \tag{6}$$

Here, d(i) represents the average intra-cluster distance between object *i* and other data points, that is, the average distance between each point within a cluster. And $c(i)$ is the average distance between the cluster where the object *i* belongs to all other clusters the object *i* does not belong, i.e. the distance between all clusters and *X* denotes the cluster.

The Silhouette Coefficients were used to determine the number of clusters. The Silhouette score yielded the same result of 6 points for the standard K-means, K-Medoids and Mini Batch K-means algorithms in the LRFMV model. In the Silhouette Coefficient calculation from Fig 3, it is spotted that the value of the Silhouette Coefficient has started decreasing after cluster 5.

Till cluster 5, it was rising, and the value was 0.344 for cluster 5. For cluster 6, the value the 0.3305, although it rose a bit in cluster 7, whereas the value was 0.3416. But gradually, the value declines and falls to 0.2956, which indicates the downfall of coefficient value.

**3.4.2 Cumulative explained variance ratio.** To lower the complexity of the suggested model LRFMV, first, reducing the dataset's dimensionality is a must. The Explained Variance Ratio is explained to determine the utility of the primary components and the number of components to include in the LRFMV model. The explained variance ratio indicates the proportion of variation that each of the specified components contributes. The cumulative percentage signifies how much variation is described by the first n component. For instance, the cumulative rate for the second component is equal to the first and second components' variance percentages. The number of components vs variance graph has been plotted in Fig 4 using this method.

Any number of components can be combined to achieve a ratio of 0.90 to 1.00 (90%-100%). Consequently, two dimensions have been chosen, demonstrating the best number of components for the proposed LRFMV model.

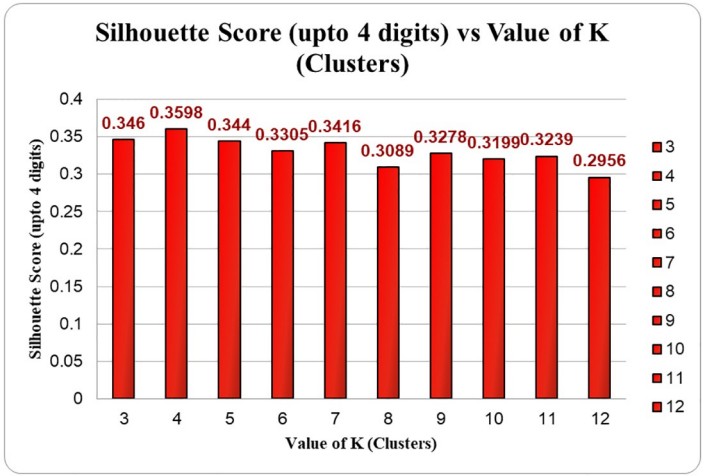

**Fig 3. Silhouette score for the LRFMV model.**

**3.4.3 PCA.** When input characteristics have a lot of dimensions and data is difficult to visualize, PCA is applied. By minimizing the space by increasing the variations, PCA also ensures that valuable information is present. It also gives the variables a synchronized low-dimensional form of the original space. Mathematically PCA aims to create a linear mapping $L$ to maximize the original data $D$ variations. In other words, PCA solves the equivalent eigen-value problem of the covariance matrix by maximizing the cost of the function $L^{T}ML$ with respect to $L$ [46] Eigenvalue. Eigenvectors are always found in pairs, and their associated Eigenvalues are equal to the data dimension. Principal components are the direction of the Eigenvectors, where Eigenvalues provide the variance from each principal component. For any given data $D$, if $d_{xy}$ demonstrates the pairwise Euclidean matrix for a high dimensional data and $\|\gamma_x - \gamma_y\|$ is the Euclidean distance between low dimensional data points $\gamma_x$ and $\gamma_y$. Afterwards, PCA will try to find the linear mapping $L$ for maximizing the cost function as

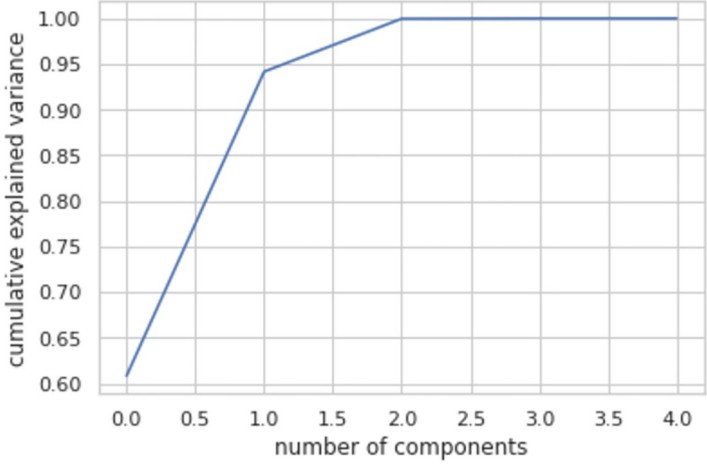

**Fig 4. Cumulative explained variance for the LRFMV model.**

below,

$$P(Y) = \sum_{x,y}(d_{xy}^2 - \|\gamma_x - \gamma_y\|^2)$$

(7)

In Fig 5, it is visible that information of 5 clusters of the LRFMV model has been represented in two dimensions by using PCA to avoid the convolution.

In Fig 6, the K-Means clustering algorithm has been applied over PCA components for the LRFMV model. It is visible that 5 clusters have been formed after applying the K-Means algorithm which has been represented by 5 distinct colors.

In the study, the features of L, R, F, M, V have been extracted, respectively, which will be complex to analyze and portray in five dimensions. The Cumulative Explained Variance Ratio is used to know how many dimensions are suitable for the dataset to represent and analyze the graphs. PCA transformed the data frame from six to two dimensions. It increased the

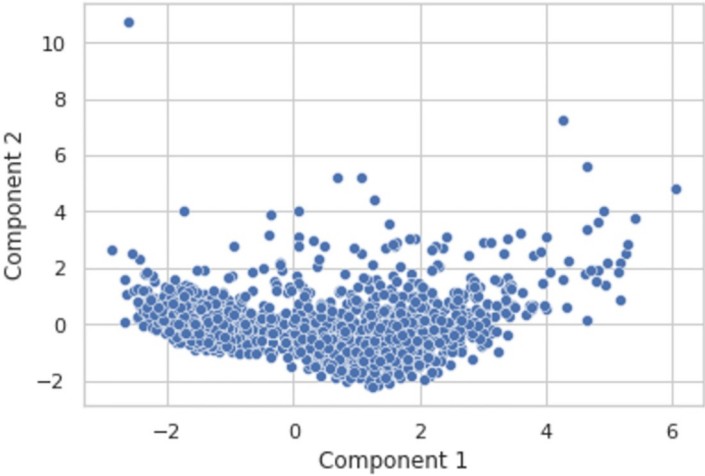

**Fig 5. PCA on the proposed dataset.**

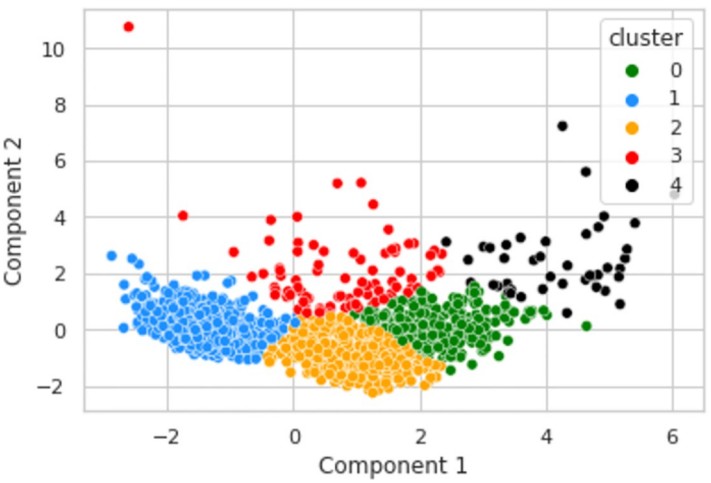

**Fig 6. K-means clusters over PCA components for LRFMV model.**

interpretability of the dataset along with reducing the number of dimensions without any information loss.

## 3.5 Correlation of volume with other features

To identify profitable customers for superstores, all six LRFMV parameters must be considered. Each parameter should have distinct characteristics that do not overlap with other attributes. Previous research demonstrated how L is associated with other attributes and why it is important to consider in order to enhance profitability in market segments [14, 35, 41, 43]. A heatmap is generated here to visualize the relationship between the newly added feature called volume and other parameters L, R, F, and M. Fig 7 represents the relationship between each parameter. The indication ranges from light to dark or strongly associated to less connected (1.0 to -0.2).

As an upgraded version of the RFM model, the Length, Recency, Frequency, and Monetary model (also known as the LRFM model) was developed to identify more precise and pertinent consumer groups for profit maximization. The properties of the current LRFM model, as seen in Fig 7, likewise demonstrate a good correlation between length and frequency (0.8) and between monetary and frequency (0.6). Even yet, many researchers utilize this strategy frequently to successfully analyze client behavior across a range of application fields and in spite of having a good correlation with frequency, length has been introduced. So we believe that volume can help to further assess how much revenue boost and marketing strategy can be developed for the superstore industry and contribute to both technical sectors and the business world.

Therefore, to validate the statement, the LRFMV model can be employed as a unique model and see if it outperforms the traditional RFM model and LRFM model.

## 4 Comparative study and result analysis

### 4.1 Comparative analysis

To determine the number of clusters for the RFM, LRFM, and LRFMV model, the Elbow Method, and the Silhouette Coefficient are applied. It is important to know the total number

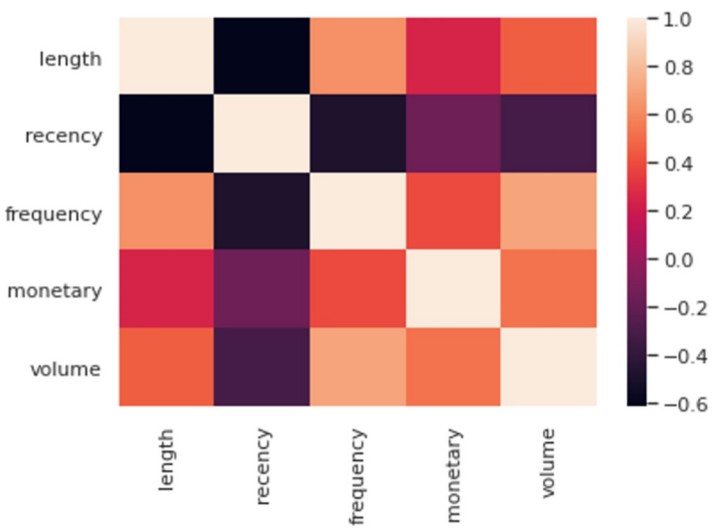

**Fig 7. Heatmap for LRFMV model.**

of clusters before applying K-means so that the data doesn't overfit in case of high value for K, and important features are not excluded in case of lower value for k. For RFM and LRFM models, 4 clusters and LRFMV, 5 clusters have been found. The quality of clustering refers to how closely the clusters fit the original data. The cluster fitness is vital because valued customers are not identified effectively by clustering if the resulting clusters do not extract the relevant attributes even after segmentation. For clustering, the conventional k-means algorithm was given higher priority due to its large and wide range of acceptance, and a comparison of profit analysis for RFM, LRFM, and LRFMV models was more successfully constructed for k-means clustering [47]. Profit per head has also been evaluated for clusters formed by the RFM, LRFM, and LRFMV models to see how well these clusters worked in K-Medoids, Mini Batch K-means, and Standard K-means algorithm.

Fig 8 illustrates the profit per head for the RFM, LRFM, and LRFMV model for each cluster adopting K-Medoids. Here cluster 0, cluster 1, and cluster 3 generate the highest profit than both RFM and LRFM models. Besides, the polarity is also observed in the proposed model, which is not visible in both RFM and LRFM models for the K-Medoids algorithm. In both RFM and LRFM models, the profits are either very low or high, but in the LRFMV model, the profit distributes from medium, high, and very high. That implies the proposed LRFMV model fits very well than RFM and LRFM models for the K-Medoids algorithm.

Moreover, the LRFMV model also has one extra cluster, which is cluster 5, to boost profitability as well as the overall profit margin is higher than others where both RFM and LRFM models do not contain the extra cluster. As a result, it can be said that the cluster quality for the LRFMV model employing the K-Medoids method is undeniably better than that of the RFM and LRFM model. Here, Fig 9 has been plotted for the Mini Batch K-means approach, and it is discernible that the difference between cluster size and earned profit is huge in LRFMV models than both RFM and LRFM models.

Comparing the profits of three different models in the above bar chart, it can be concluded that the clusters of the LRFMV model are making the highest profit in cluster 0, cluster 1, and cluster 2 than LRFM and RFM models, which are denoted with green bars. However, the polarity is less for the three models varying from very low, medium, and high. As the cluster number is larger here, the clustering quality and behavior of particles in each cluster are more similar. The largest cluster for the LRFMV model is cluster 1, and earned profit is 2071.9777, and the smallest one is cluster 4, where the profit is 76.712009. Therefore, the proposed

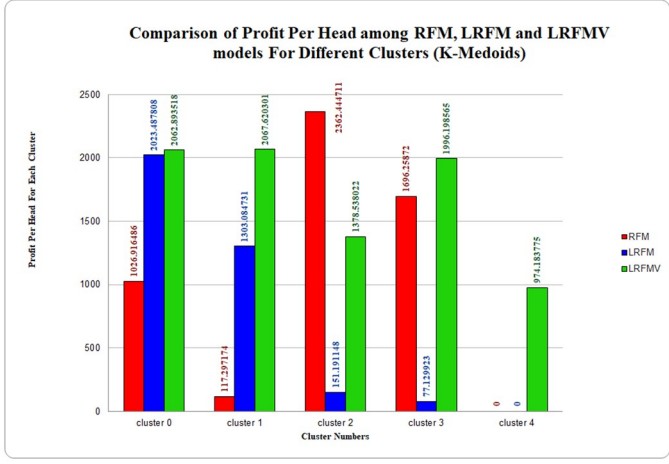

**Fig 8. Profit analysis for RFM, LRFM and LRFMV models using K-Medoids algorithm.**

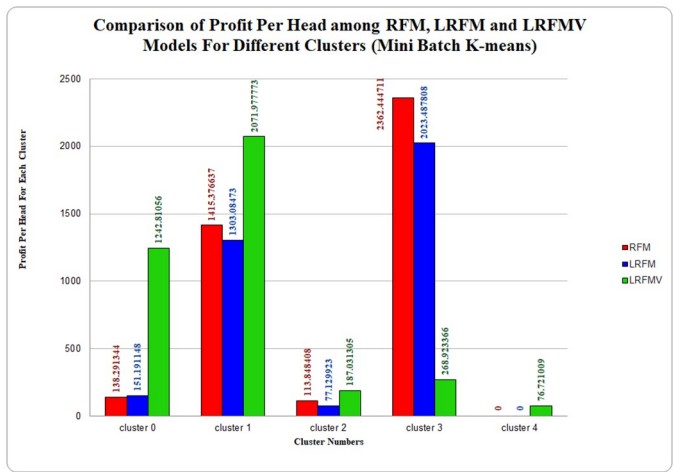

**Fig 9. Profit analysis for RFM, LRFM and LRFMV model using Mini Batch K-means algorithm.**

LRFMV model is undeniably better than both RFM and LRFM models for the Mini Batch K-means algorithm.

Now, in Fig 10 with the results from the RFM model, it is shown that there are four groups, each with its per-head total revenue to the superstore for the K-means algorithm. The highest profit is achieved from cluster 3, which is about 2362.444711, while the lowest profit is achieved from cluster 0, which is about 110.809001. For the LRFM model, cluster 3 is the smallest, 39.509398, and cluster 1 is the highest profit-making cluster which is 1642.51772. Afterward, in the graph of the LRFMV model, it is visible that the LRFMV model can develop four segments with the same number of customers while providing more variation of profit per cluster. The highest profit is achieved from cluster 1 with 1640.468891 profit per head. The lowest profit is achieved from cluster 4 that is -5.153294 per head. Although, in this algorithm, RFM provided the largest profit than both LRFM and LRFMV in one cluster, and LRFM provided the largest profit than both RFM and LRFMV in two clusters with one a very little difference than LRFMV model, it is observed special characteristics in the LRFMV model.

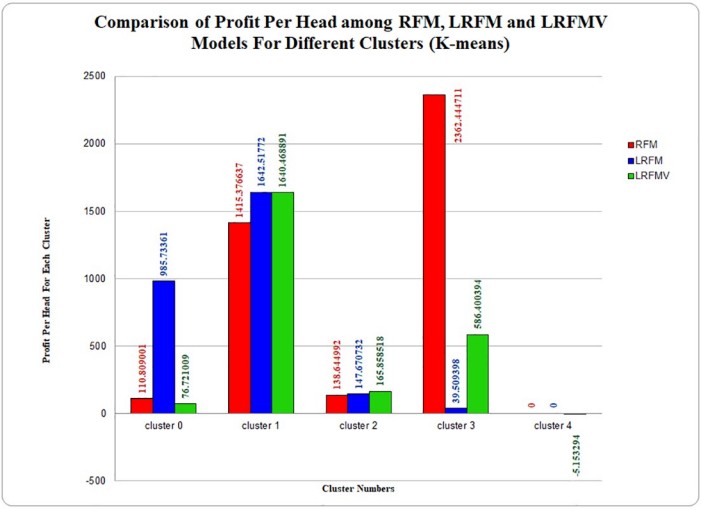

**Fig 10. Profit analysis for RFM, LRFM and LRFMV models using standard K-means algorithm.**

From comparative analysis, we can see that the LRFMV model is providing a reasonable output compared to the RFM and LRFM models and is ensuring the polarity, which is not present in RFM or LRFM models for any of the three clustering algorithms. We suggested volume, which was determined by assessing the typical number of items a regular superstore customer purchases in a single day. The previously discovered average quantity was divided by the entire number of days in the customer's brief period of visitation in order to obtain the final average as the volume in a given time frame. There are some expensive products in the superstores as well as very cheap products. However, the price of the most common items frequently bought by the customers are usually daily groceries and do not vary much [48–50]. Additionally, the LRFMV model has one additional cluster while using the K-Medoids algorithm, to increase profitability and the overall profit margin is higher than other models, where both RFM and LRFM models do not have the additional cluster, which also guarantees superior clustering quality. In the bar chart of Fig 9, which compares the profitability of three distinct models, it can be seen that the LRFMV model's cluster which is represented by the green bars makes the most profits for clusters 0, 1, and 2 compared to LRFM and RFM models for Mini batch K-Means Algorithm. Furthermore, while the LRFMV model uses K-Means Algorithm to produce a structured output, RFM and LRFM models reveal conflicting profit analyses for various clusters. We can assert that this has occurred in order to add volume to the existing models because the LRFMV model is producing a passable output when compared to the RFM and LRFMV models. Most business models try to find only profit but sometimes ignore the fact that losses can be dreadful for the company. With the proposed LRFMV model, cluster 4 identifies the customers who give only losses without any profits. By this, the proposed LRFMV model indicates a very important role in boosting profit margin and decreasing the loss percentage of a superstore. Therefore, in many ways, the clustering quality of the LRFMV model is preferable for the K-means algorithm as well.

## 4.2 Result analysis

The volume-profit analysis calculates the impact of changes in a company's sales volume and price on profit. This paper aimed to initiate a new term volume (V) with the existing LRFM model to ensure better clustering and to find out potential, valuable and profitable customers. The volume and profit have a proportional relationship with each other. It will be typified that the customer cluster with a large volume ensures more profit than others. Regardless of the expenditure, customers with more buying habits directly influence the profit of any firm. Later, it will be proven by different charts and graphs where it is noticed that the more customers buy products, the larger profit the superstore can make for K-Medoids, Mini Batch K-means, and K-means algorithms.

To understand the volume-profit relationship for the K-Medoids approach on the proposed LRFMV model, a bar chart has been plotted in Fig 11. It is observable that cluster 1 is giving the lowest profit, which is 151.233421, where the volume of the sold product is also the lowest, which is 2.180618. On the other hand, cluster 4 gives the highest amount of profit which is 2062.893518, and the volume of the sold product is also maximum that is 3.951606.

Actual objects can be used to represent clusters in the K-Medoids methodology, with one representative item per cluster, and a medoid is an object in a cluster with the minimum average dissimilarity to all other objects in the cluster. The other objects are grouped along with the most similar representative object. A proportional relationship between volume and profit is noticeable for this approach as well. Its features include the ability to solve K-means challenges and be robust to outliers or interference. By introducing a volume feature with this algorithm will make the most profits in superstores.

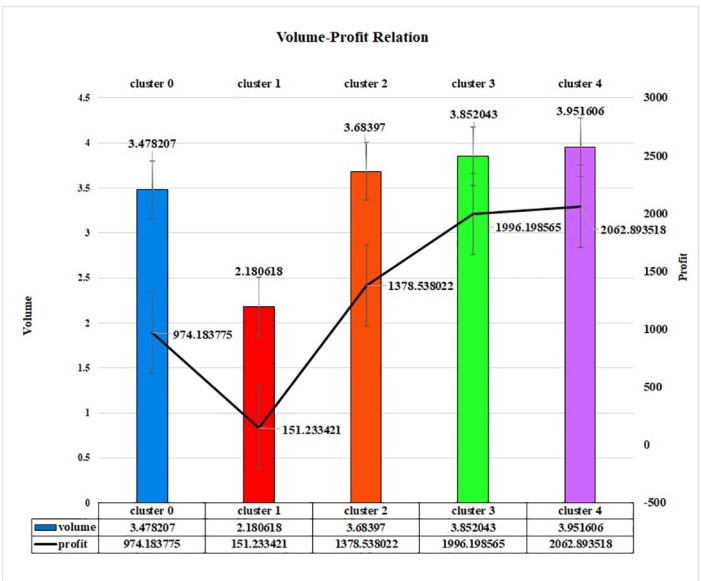

**Fig 11. Volume-profit relationship of LRFMV model for K-Medoids algorithm.**

It is evident that a proportional relationship is also present in Fig 12 for the Mini Batch K-Means approach. The highest profit has been earned from cluster 4, which is 2071.977773, and the volume is also the highest, which is 3.913318. Similarly, the profit of cluster 2 is 76.721009, where the volume is also the lowest, which is 2.030807. Hence, a direct relationship between volume and profit indicates the importance of the proposed newly introduced LRFMV model.

By sampling a fixed-size subsample of the dataset rather than the entire dataset, this algorithm saves time. The fundamental concept is to use tiny stochastic groups of adjusted data

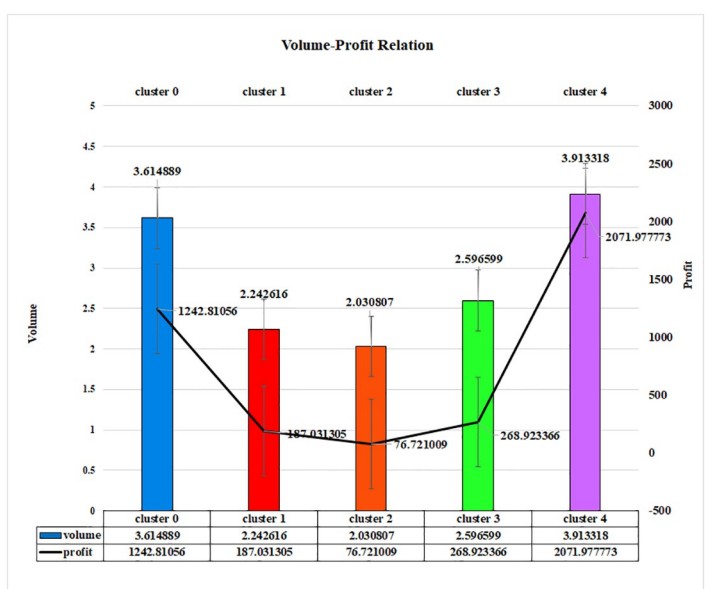

**Fig 12. Volume-profit relationship of LRFMV model for Mini Batch K-Means Algorithm.**

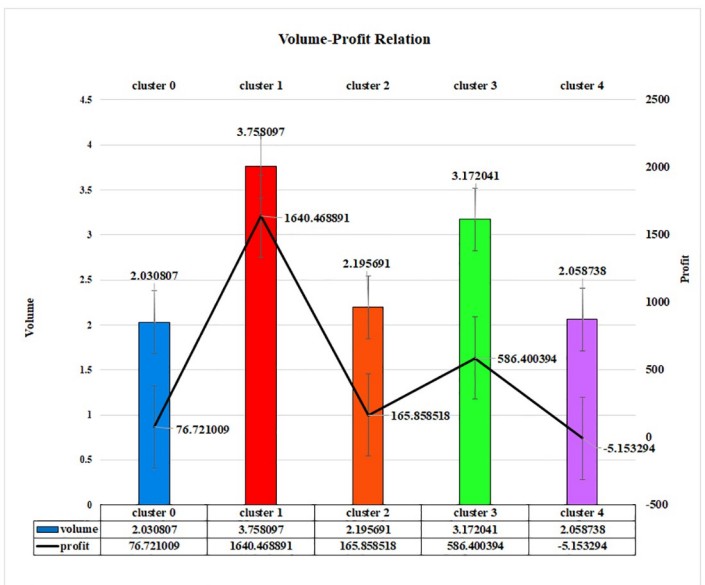

**Fig 13. Volume-profit relationship of LRFMV model for K-Means algorithm.**

that may be kept in storage. As the number of iterations grows, each mini-batch refreshes the clusters with a convex mix of prototype and example values and a falling learning rate. The number of examples assigned to a cluster is inversely proportional to the speed with which the procedure is completed. The influence of additional instances lessens as the number of iterations increases, and convergence can be detected when no changes in the clusters occur over a long period of time. Besides, a clear relationship between volume and profit is identified through this model. By this, collaborating with the proposed LRFMV model, the Mini Batch K-Means algorithm is able to identify more clusters generating higher revenue per cluster for superstores.

Here in Fig 13, a relation between Volume-Profit is shown for the K-Means algorithm, and each cluster represents customers' purchasing volume. Volume refers to the number of purchased products by different types of the customer on a single transaction. The highest amount of profit was generated through the highest volume of customers. Here, cluster 1 has generated the highest profit compared to other clusters. Customers of cluster 1 have purchased the highest volume of products, and it is about 3.758097, and the profit value is 1640.468891. Cluster 3 has generated the second-highest profit. The volume of cluster 3 is 3.172041, and the amount of profit shows 586.400394, and cluster 4 is the lowest profitable where the profit is -5.153294, and the mean volume is 2.058738. The addition of volume has been able to showcase the amount of profit which clearly contributes widely to the segmentation of the market. These numerical values clearly indicate that the volume and profit of that super shop are proportionally dependent on each other as profit is higher when the volume is comparatively greater. Moreover, when the volume is less, the loss is also persistent in the case of cluster 4.

To establish and explore new business opportunities in today's world, it has become essential to mix technical innovation and knowledge with business. In light of this, the goal of this article was to provide a novel technology element that would enable an organization to increase its profitability. In order to find the consumer group that would guarantee and provide a larger profit to a business, a new feature V has been introduced to the existing LRFM model. It can be noticed clearly that the profit-volume relationship is proportionate, as shown

by the preceding figures (Figs 11–13), and any business can generate a sizable profit by analyzing the volume, which is a scaled-down representation of the quantity of things purchased by a potential consumer. By emphasizing the amount of product those clients have purchased over a predetermined time frame or a predetermined number of visits, it helps any business identify a group of valuable customers.

This volume-profit analysis is used to determine how variations in the amount purchased influence a superstore's profit. Businesses can use this study to determine how many units they need to sell to break even (cover all costs) or achieve a given profit margin. Because of its widespread use, the K-means algorithm has been used for further explanation in this paper. The profit earned from the clusters using the K-means algorithm has been plotted in the bar chart below, where the x-axis reflects the cluster number and the y-axis indicates the profit earned.

In the bar chart of Fig 14, Cluster 1 is the most profitable cluster consisting of total profits of 1371432, while Cluster 4 is the least profitable with total profits of -262.818. Moreover, cluster 1 generates almost more than 90% profits than all other clusters, and by this most profitable cluster can be identified. Besides, cluster 4 actually generating losses which can be a hindrance to boosting the superstore business.

In Fig 15, it is observed the distribution of customers in each cluster for the LRFMV model with the K-means algorithm. In Fig 13, cluster 1 yields the most profit, and from the below figure, the percentage of customers in this cluster is also the highest (51.57%). Here, cluster 0 and cluster 2 contains a decent percentage of customers (14.19% and 25.66%) yet yields very low profit for the superstore. Moreover, almost 3.15% of customers in cluster 4 generate losses for the superstore business, which can be addressed easily with the proposed LRFMV model. In the Customer Classification Matrix, the Aggressive Customer type was made up of this cluster.

Several literature positions are advised in Table 6 using the Customer Classification Matrix, which demonstrates that businesses can serve lucrative clients in various ways. The most valuable clients are those who are **passive**, providing great revenue at low cost. These are the most profitable customers, and the corporation should pay them special attention. Some customers

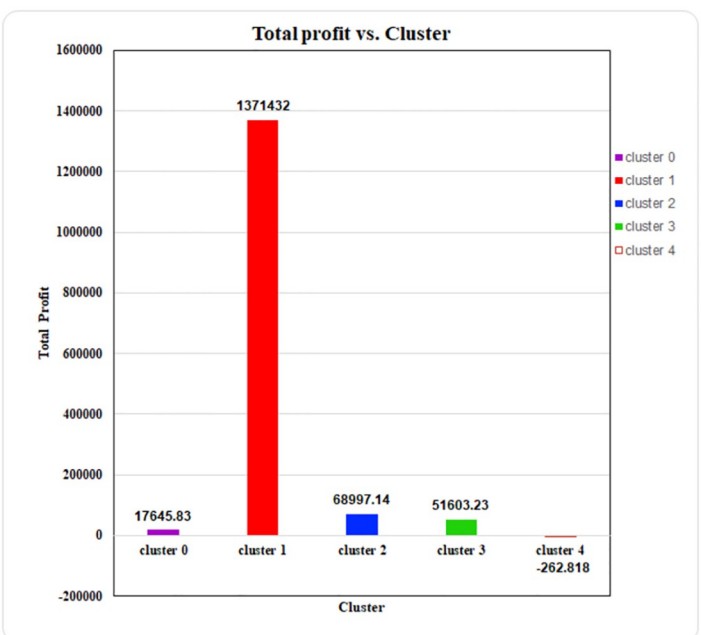

**Fig 14. Profit of each cluster in LRFMV model.**

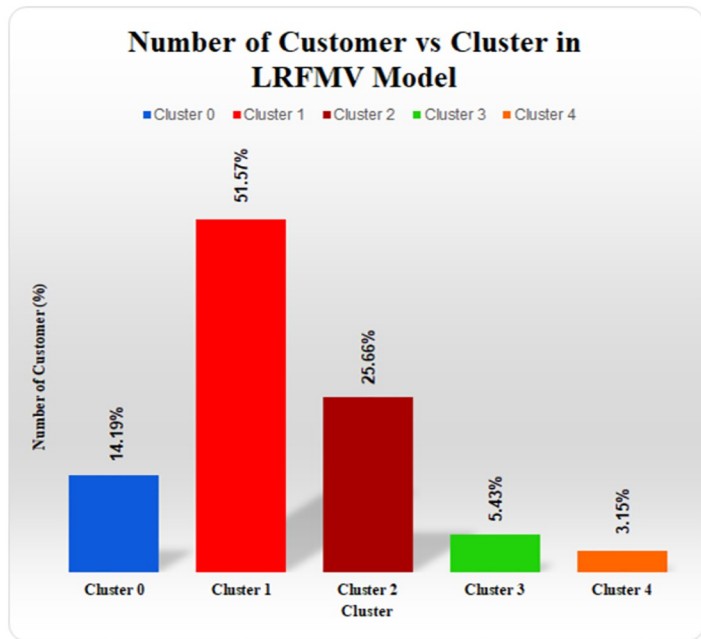

**Fig 15. Number of customers (%) in each cluster for LRFMV model.**

that generate a lot of money can also be expensive, known as **carriage trade**- they can be lucrative if the revenue surpasses the expense of serving them. There may be customers who are easy to satisfy yet don't bring in a lot of money, and they are known as **bargain basement**. Finally, clients with high costs and poor revenue are included in the last quadrant named **aggressive**.

For LRFMV modelling, a pre-processed data set is used, and the average of each component of LRFMV was calculated for the whole dataset. The LRFMV components for six separate clusters are also calculated, as well as the profit at this point. The calculated averages are listed, and the values of L, R, F, M, V for each of the six clusters were compared with those averages. Giving numerical values to simplify the representation is avoided here, and Table 7 is created for easier understanding.

Here, "Cluster" indicates the cluster number (0,1,2,3,4). After comparing, if a higher value has been obtained than the average, it was written up, and if a lower value is obtained for each

**Table 6. Revenue generation for each cluster and their cost to serve.**

| **Passive** (Revenue high, cost to serve low) | **Carriage Trade** (Revenue high, cost to serve high) |
|---|---|
| **Bargain Basement** (Revenue low, cost to serve low) | **Aggressive** (Revenue low, cost to serve high) |

**Table 7. Comparison amongst the avg of L, R, F, M, V with each cluster.**

| Cluster | L | R | F | M | V | Profit | Customer Type |
|---|---|---|---|---|---|---|---|
| 0 | ↓ | ↑ | ↓ | ↓ | ↓ | Incredibly low | Bargain Basement |
| 1 | ↑ | ↓ | ↑ | ↑ | ↑ | Incredibly high | Passive |
| 2 | ↑ | ↓ | ↓ | ↓ | ↓ | low | Bargain Basement |
| 3 | ↓ | ↑ | ↓ | ↑ | ↑ | Average | Carriage Trade |
| 4 | ↓ | ↑ | ↓ | ↓ | ↓ | Negative | Aggressive |

component (L, R, F, M, V), it was denoted as down. The profit of different clusters was also compared and stated a comparison with the up ("↑") / down ("↓") method. The last column describes the customer type based on "L", "R", "F", "M" and "V" indications. It helps to get a better idea of different types of customers for divergent clusters.

The goal of this paper is to bring business and technology together to boost profits for superstores. Customer profitability measurement is critical for long-term business performance since it allows us to observe if particular clients are costing us money rather than making money. It becomes simple to examine client profitability as much as an organization requires once it has a structure in place to measure it. A consumer segment can be proved less important to the business than others which were previously considered profitable before analyzing them by using any metrics.

These results from different customer profitability metrics can then be used to develop and shift business strategies in order to maintain business objectives and goals on track. As an enhanced business model LRFMV, is proposed in this research to locate successful customer segments for the superstore business, it is necessary to evaluate customer segments which have been created by using different clustering algorithms by using various business metrics to offer organization management with an awareness of each customer profitability. Among different customer Customer Profitability Analysis, strategy alignment and customer profitability matrix is one of the most important measuring tools for grouping information into customer profitability segments which allows companies to take different, targeted actions and strategies against different profitability segments with the goal of increasing the company's total profitability.

In Table 8, all consumers are classified into four groups, each with its own method for dealing with customers.

Profitable clients in the **"Target"** group are linked to the action **"RETAIN"** and the company should investigate the prospect of expanding commercial connections with such customers, as long as the business model does not change considerably.

Profitable customers in the **"Non-target"** group are linked to the act **"MONITOR"** and these consumers must be regularly watched to avoid falling into the **"Non-target"** and unprofitable sector. Unprofitable customers in the **"Target"** group as related to the action **"TRANSFORM"** and the corporation should use various techniques to convert these customers into profitable segments, or at the very least, bring them to a cut off point. Depending on the company-customer business circumstances, different tactics will be employed.

Unprofitable clients in the **"Non-target"** category are linked to the action **"REPLACE"** to whom the company should stop investing in their development. The suggested answer is to raise product or service selling prices until the consumers fall into the **"MONITOR"** sector or move their business to another provider. If this occurs, the corporation may be able to refocus its efforts toward serving the most profitable customers.

If we utilize strategy alignment and customer profitability matrix to evaluate our findings of 5 separate clusters of customers via centroid-based algorithms, we will get the following results. Out of the five client clusters, the first four are lucrative, while the fifth, cluster 4, is non-profitable with a negative value of -262.818. Cluster 1 is the most profitable client segment, with a

**Table 8. Analysis of profit for each cluster and identifying potential customer segment using customer profitability matrix.**

| Retain (Profitable, Target) | Transform (Unprofitable, Target) |
|---|---|
| Monitor (Profitable, Non-target) | Replace (Unprofitable, Non-target) |

**Table 9. Customer type analysis and target audience identification for each cluster.**

| Cluster | Profit | Target Audience | Customer Type |
|---------|--------|-----------------|---------------|
| Cluster 0 | 17645.83 | No | Monitor |
| Cluster 1 | 1371432 | Yes | Retain |
| Cluster 2 | 68997.14 | No | Monitor |
| Cluster 3 | 51603.23 | Yes | Retain |
| Cluster 4 | -262.818 | No | Replace |

total profit of 1371432. Clusters 1 and 3 are considered successful since they generate a profit greater than their average profit, as seen in Table 9.

This matrix boosts profits by removing unprofitable consumers and increasing sales or services to profitable ones. It is an assessment of the true costs of each client group, including taking non-production expenses into account when calculating profitability, and it indicates that non-production costs can occasionally surpass production costs.

## 5 Conclusion

Numerous studies on customer segmentation using RFM and LRFM models have been performed since the invention of these concepts. Still, only a few of them can create a connection between customers and commodity quantity. The proposed analysis makes a significant impact by establishing a strong correlation between earnings per head and the commodity purchased by each customer in a single transaction. The research demonstrates a novel method for segmenting customers into productive clusters based on the volume of the commodity. The model presented in LRFMV research can resolve a variety of issues related to determining the optimal customer for the optimal product.

One of the more popular uses of K-means clustering is segmenting customers to get a greater view of them, which can then maximize sales. Similarly, customer segmentation is a technique for improving contact with customers and learning about their desires and activities so that companies' issues can be developed. Customer segmentation is essential to acclimate new customers and maximize earnings. Prospective customer data may be used to deliver programs depending on the type of customers, such as internet advertising, purchasing and sale. Additionally, the objective of K-means is to classify data points into separate, non-overlapping sub-populations.

Along with K-Means, other algorithms like K-Medoids, Mini Batch have also been used to cross-check the K-means clusters for this dataset. The LRFMV analysis will assist company owners in a more efficient way for segmenting their clients, which would result in more efficient contact. Moreover, the proposed LRFMV approach can be applied to databases of non-discrete details and a smaller variation in the number of data points. Additionally, since k-means clustering is prone to outliers, it is preferable to exclude them first.

Numerous companies could potentially use this model in the future to derive market characteristics from matrices of customer research. Similarly, this technique can be used for datasets that have a low variance in terms of commodities. Along with K-means, other clustering techniques were adopted in the study, such as K-Medoids, which are particularly useful for dynamic datasets and Mini Batch K-means, which can be used to save time and space. Additionally, this model can be used in future to analyze other aspects of advertising, and its reliability can be quantified using certain matrices.

## Author Contributions

**Conceptualization:** Shahnur Azad Chowdhury, Md. Golam Rabiul Alam.

**Data curation:** Rezwana Mahfuza, Nafisa Islam, Md. Toyeb, Md Asaduzzaman Faisal Emon.

**Formal analysis:** Rezwana Mahfuza, Nafisa Islam, Md. Toyeb, Md Asaduzzaman Faisal Emon.

**Investigation:** Shahnur Azad Chowdhury, Md. Golam Rabiul Alam.

**Methodology:** Rezwana Mahfuza, Nafisa Islam, Md. Toyeb, Md Asaduzzaman Faisal Emon, Shahnur Azad Chowdhury, Md. Golam Rabiul Alam.

**Project administration:** Shahnur Azad Chowdhury, Md. Golam Rabiul Alam.

**Resources:** Shahnur Azad Chowdhury, Md. Golam Rabiul Alam.

**Software:** Rezwana Mahfuza, Nafisa Islam, Md. Toyeb, Md Asaduzzaman Faisal Emon.

**Supervision:** Md. Golam Rabiul Alam.

**Validation:** Rezwana Mahfuza, Nafisa Islam, Md. Toyeb, Md Asaduzzaman Faisal Emon, Shahnur Azad Chowdhury.

**Visualization:** Rezwana Mahfuza, Nafisa Islam, Md. Toyeb, Md Asaduzzaman Faisal Emon.

**Writing – original draft:** Rezwana Mahfuza, Nafisa Islam, Md. Toyeb, Md Asaduzzaman Faisal Emon.

**Writing – review & editing:** Shahnur Azad Chowdhury, Md. Golam Rabiul Alam.

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
