## [Decision Letter · Decision Letter 0]

4 Apr 2022

PONE-D-21-32764LRFMV: An Efficient Customer Segmentation Model for SuperstoresPLOS ONE

Dear Dr. Alam,

Thank you for submitting your manuscript to PLOS ONE. After careful consideration, we feel that it has merit but does not fully meet PLOS ONE’s publication criteria as it currently stands. Therefore, we invite you to submit a revised version of the manuscript that addresses the points raised during the review process.

There are some issues raised by the reviewers. I am thus recommending to revise the paper before taking final decision.

We look forward to receiving your revised manuscript.

Kind regards,

Siuly Siuly, PhD

Academic Editor

PLOS ONE

https://journals.plos.org/plosone/s/file?id=ba62/PLOSOne_formatting_sample_title_authors_affiliations.pdf".

2. Please clarify in your Methods section whether the customer names and other identifying characteristics in the dataset used are fictional.

Reviewers' comments:

Reviewer's Responses to Questions

**Comments to the Author**

1. Is the manuscript technically sound, and do the data support the conclusions?

Reviewer #1: Yes

Reviewer #2: Yes

2. Has the statistical analysis been performed appropriately and rigorously? 

Reviewer #1: Yes

Reviewer #2: Yes

3. Have the authors made all data underlying the findings in their manuscript fully available?

Reviewer #1: Yes

Reviewer #2: Yes

4. Is the manuscript presented in an intelligible fashion and written in standard English?

Reviewer #1: Yes

Reviewer #2: Yes

5. Review Comments to the Author

Reviewer #1: Authors presented a customer segmentation model, namely LRFMV (Length, Recency, Frequency, Monetary and Volume) and studied the profit-quantity relationship. The paper is thoroughly written, and the results are convincing. My minor comments are the following:

1. The data obtained from the LRMFV model is clustered by using conventional K-means, K-Medoids, and Mini Batch K-means methods. Provide some rationalities why these models have been chosen for this clustering?

2. Please use labels for Figures 7-9

3. Please provide a paragraph about the technical contributions of this research study. Section 1.2 seems summarizing the benefits of the proposed method.

Reviewer #2: In PONE-D-21-32764, Alam et al. presented a model for analyzing client segments and profit.

S1. They extended existing RFM and LRFM models by adding volume to form a LRFMV model.

W1. The Author Summary can be improved.

W2. It is unclear what are significant benefits of incorporating volume in exising RFM or LRFM models.

W3. More exhaustive evaluation would be helpful.

W4. Legend for Fig. 5 needs to be updated. Currently this figure is meaningless without proper bar labels.

6. PLOS authors have the option to publish the peer review history of their article (what does this mean?). If published, this will include your full peer review and any attached files.

Reviewer #1: No

Reviewer #2: No

---

## [Author Response · Author response to Decision Letter 0]

23 Jun 2022

RESPONSE TO REVIEWER #1’S COMMENTS

Reviewer#1, Concern # 1: The data obtained from the LRMFV model is clustered by using conventional K-means, K-Medoids, and Mini Batch K-means methods. Provide some rationalities why these models have been chosen for this clustering?

Author response: We would like to express our deepest gratitude to the honorable reviewer for the insightful comment and we are much obliged for the valuable suggestions. Many clustering methods are used for grouping similar data objects inside the same set based on similarity criteria, such as density-based, centroid-based, and hierarchical clustering. Density-based algorithms link core objects and their surroundings to generate dense regions known as clusters, which are asymmetrical and made up of the most densely connected points. Furthermore, the hierarchical clustering method merges or separates related data objects by creating a hierarchy of clusters, also known as a dendrogram, where the same objects can reside in multiple clusters. By hypothesizing a model for each cluster, model-based clustering techniques discover the best fit of data for a particular model. Grid-based algorithms use dense grids to generate clusters and use multi-resolution grid data structures. It quantizes the original data space into a finite number of cells that constitute the grid structure before performing all operations on the quantized space. However, the key objective of this research is to partition the customers based on L, R, F, M, and V without overlapping the cluster elements. Therefore, we have chosen partition-based clustering methods i.e., K-Means, Mini-batch K-Means, and K-Medoids for the customer segmentation. 

For the large and diversified data set like our superstore dataset, centroid-based clustering like K-Means was the preferred one as it is fastest and scalable. The Mini-batch K-Means can reduce the time it takes to compute a growing dataset. However, K-Means and Mini-batch K-Means algorithms are sensitive to outliers. The K-Medoid mitigates sensitivity to outliers whilst still ensuring convergence. 

Author action: Updated sections have been marked yellow using the highlighter tool in the revised

manuscript.

Modifications made to the manuscript in this regard is as follows:

[Updated] [Section: 1.1 Background and Motivation]

Reviewer#1, Concern # 2: Please use labels for Figures 7-9

Author response: We would like to offer our sincere regrets for overlooking these issues and express our deepest gratitude to the honorable reviewer for specifying these through insightful comments. As the respected reviewer indicated, we have labeled the axes of Figures 7-9 in the updated manuscript. However, as an extra image (Figure 7) has been added in the updated manuscript, the Figures 7, Figure 8 and Figure 9 in the previous paper have been termed as Figure 8, Figure 9 and Figure 10.

Author action: Updated axes’ labels of figures have been marked yellow using the highlighter tool in the revised manuscript.

Modifications made to the manuscript in this regard is as follows:

[Updated] [Section: 4.1 Comparative Analysis]

Reviewer#1, Concern # 3: Please provide a paragraph about the technical contributions of this research study. Section 1.2 seems to summarize the benefits of the proposed method.

Author response: We would like to convey our heartfelt gratitude to the esteemed reviewer for his insightful comment. As recommended by the respected reviewer, we have added a paragraph about the technical contributions of this research in the introduction section of the revised manuscript. 

Author action: Updated sections have been marked yellow using the highlighter tool in the revised

manuscript.

Modifications made to the manuscript in this regard is as follows:

[Updated] [Section: 1.2 Contributions]

RESPONSE TO REVIEWER #2’S COMMENTS

Reviewer#2, Concern # 1: The Author Summary can be improved.

Author response: We would like to express our deepest gratitude to the honorable reviewer for the insightful comment and we are much obliged for the valuable suggestions. Having said that, we have revised the Author Summary and updated it.

Author action: Updated sections have been marked yellow using the highlighter tool in the revised

manuscript.

Modifications made to the manuscript in this regard is as follows:

[Updated] [Author Summary]

Reviewer#2, Concern # 2: It is unclear what are significant benefits of incorporating volume in existing RFM or LRFM models.

Author response: Thank you for the astute comment and recommendation. We have incorporated how adding volume to existing RFM or LRFM models can be beneficial in both the technical and business worlds in the revised manuscript. 

Author action: Updated sections have been marked yellow using the highlighter tool in the revised

manuscript.

Modifications made to the manuscript in this regard is as follows:

[Updated] [Section 3.3.5: Computation of V]

Reviewer#2, Concern # 3: More exhaustive evaluation would be helpful.

Author response: Thank you for your valuable suggestions. As per the respected reviewer’s recommendation, we have appended a customer profitability matrix for better inception of the research. 

Author action: Updated sections have been marked yellow using the highlighter tool in the revised

manuscript.

Modifications made to the manuscript in this regard is as follows:

[Updated] [Section: 4.2 Result Analysis]

Reviewer#2, Concern # 4: Legend for Fig. 5 needs to be updated. Currently this figure is meaningless without proper bar labels.

Author response: We would like to thank the honorable reviewer for the intuitive comment and suggestion. We have incorporated your recommendation in the revised version of this manuscript. For better explanation, instead of a bar chart we have included the values of two-dimensional principal components for the LRFMV model with scatter plots in Figure 5. Besides, we have also added a depiction of K-Means clusters over PCA components for the LRFMV model. To facilitate visualization, the clusters are labeled with six distinct hues.

Author action: Modifications made to the manuscript in this regard is as follows:

[Updated] [Section: 3.4.3 PCA]

---

## [Decision Letter · Decision Letter 1]

12 Oct 2022

PONE-D-21-32764R1LRFMV: An Efficient Customer Segmentation Model for SuperstoresPLOS ONE

Dear Dr. Golam Rabiul Alam,

Thank you for submitting your manuscript to PLOS ONE. After careful consideration, we feel that it has merit but does not fully meet PLOS ONE’s publication criteria as it currently stands. Therefore, we invite you to submit a revised version of the manuscript that addresses the points raised during the review process.

ACADEMIC EDITOR:The paper received two major revisions, one in acceptance of the document and one in rejection. Therefore I decided to accept the article with major revisions.

You are now asked to follow the comments at the bottom to improve and integrate the work and send it back by the date indicated.

We look forward to receiving your revised manuscript.

Kind regards,

Vincenzo Basile, PhD

Academic Editor

PLOS ONE

Reviewers' comments:

Reviewer's Responses to Questions

**Comments to the Author**

1. If the authors have adequately addressed your comments raised in a previous round of review and you feel that this manuscript is now acceptable for publication, you may indicate that here to bypass the “Comments to the Author” section, enter your conflict of interest statement in the “Confidential to Editor” section, and submit your "Accept" recommendation.

Reviewer #1: All comments have been addressed

Reviewer #3: (No Response)

2. Is the manuscript technically sound, and do the data support the conclusions?

Reviewer #1: Yes

Reviewer #3: No

3. Has the statistical analysis been performed appropriately and rigorously? 

Reviewer #1: Yes

Reviewer #3: No

4. Have the authors made all data underlying the findings in their manuscript fully available?

Reviewer #1: Yes

Reviewer #3: Yes

5. Is the manuscript presented in an intelligible fashion and written in standard English?

Reviewer #1: Yes

Reviewer #3: Yes

6. Review Comments to the Author

Reviewer #1: Authors have addressed all my comments in the revised manuscript. I don't have any further comments on this manuscript.

Reviewer #3: This study proposed a new LRFMV by incorporating A new dimension V (volume) to existing LRFM model to effectively segment customers in t superstores. This new feature, volume (V) is the purchased quantity and is defined as the average quantity of products purchased by each customer.

With the features of the classical LRFM model, it is possible to identify customer groups, such as profitable, non-profitable, potential, uncertain, loyal, new, and etc. Thus this model is utilized by many researchers in different application domains to analyze customer behaviors, to identify different and target customer groups.

Considering these, how will this new feature, the purchased quantity contribute to the modelling of customer behaviour? Are there any studies showing the effect of this feature on modelling customer behavior or customer profitability? Is there any study showing the direct relationship between profit and quantity ? The study lacks literature supporting the profit-amount relationship on which the proposed model is based.

Moreover, the authors provides Fig 7 to represent the relationship between each parameter in the model. Authors assert that the newly introduced parameter volume (V) has a very weak relationship with each of the other parameters and is quite strong with itself. Thus, they conclude that V is a unique feature and can not be replaced by other features. However, when we observe that figure, it is clearly seen that new proposed feature volume (V) directly correlates with both frequency (around 0.8) and monetary (around 0.6). These values are quite high for the correlation analysis. When this is taken into account, it is seen that the new feature does not add any extra dimension to the current model.

Authors state that the volume highlights the customers who provide more profit to the organization. However, customer profitability is not related to the amount purchased. Monetary (M) and frequency (F) variables explain customer profitability in the standard model. Let's try to explain with an example that the amount purchased does not add an extra dimension and is a meaningless variable in terms of customer segmentation. There is a customer and who buys a lot of products, but that customer may only buy certain products from the relevant supermarket that are quite cheap. In other words, the total amount paid is quite less. In this case, when we consider the amount received, how can we call this customer profitable? Or let's think the opposite, let's say that another customer who buys very few products, and the products he buys are quite expensive. In other words, the total amount paid is quite high. In this case, when we consider the amount received, how can we call this customer non-profitable? Therefore, purchased quantity as seen in the example, does not offer an extra dimension in explaining customer behaviour.

In summary, considering my above concerns, the contribution and technical sound of this paper is not suitable to be published in such a journal.

7. PLOS authors have the option to publish the peer review history of their article (what does this mean?). If published, this will include your full peer review and any attached files.

Reviewer #1: No

Reviewer #3: No

---

## [Author Response · Author response to Decision Letter 1]

28 Nov 2022

RESPONSE TO REVIEWER #1’S COMMENTS

Reviewer#1, Comment: Authors have addressed all my comments in the revised manuscript. I don't have any further comments on this manuscript.

Author response: We want to express our sincerest appreciation to the respected reviewer for the reviews and feedback given in the previous round of review. Your comments and reviews surely help us to improve the quality of our research work and the manuscript as well. Thank you for your positive comment on our research work.

RESPONSE TO REVIEWER #3’S COMMENTS

Reviewer#3, Concern # 1: With the features of the classical LRFM model, it is possible to identify customer groups, such as profitable, non-profitable, potential, uncertain, loyal, new, and etc. Thus this model is utilized by many researchers in different application domains to analyze customer behaviors, to identify different and target customer groups. Considering these, how will this new feature, the purchased quantity contribute to the modelling of customer behaviour? Are there any studies showing the effect of this feature on modelling customer behavior or customer profitability? Is there any study showing the direct relationship between profit and quantity? The study lacks literature supporting the profit-amount relationship on which the proposed model is based.

Author response: We want to express our sincere appreciation to the respected reviewer for his thoughtful feedback. In the Literature Review section, we have included some studies based on the profit-quantity relationship. Moreover, in the result analysis section of the revised article, we have added a paragraph discussing the profit-volume relationship of this research as suggested by the esteemed reviewer.

In this research article, we suggest a term that enhances the LRFM model by identifying the customer clusters that increase an organization's profitability. It demonstrates that regardless of the amount of money spent, a consumer will contribute more to the organization's profit if he purchases a significant amount of goods throughout specific trips. The possibility of maximum profit increases with the number of products a customer purchases. The following part, which can be found in section 4.2 known as result analysis, reveals that the majority of the sectors' profits increase in direct proportion to sales volume. In the case of superstores, there is a clear correlation between profit and volume, which may be key to implementing some efficient business strategies. We have shown the profit-volume relationship in bar charts for the three clustering methods utilized in this paper, K-Medoids, Mini Batch K-means, and K-means, in figures 11 to 13.

Author action: Updated sections have been marked yellow using the highlighter tool in the revised

manuscript.

Modifications made to the manuscript in this regard is as follows:

[Updated] [Section: 2 Literature Review]

Over the years, many data mining techniques have been widely applied in a variety of fields. As the volume of transactions in the business grows, it has become more difficult to segment profitable customers to enhance sales. "Quantity" has been incorporated in different studies for profit maximization as well as association rule mining [44] [45]. The volume or quantity produced directly relates to the revenue, cost, and profit function2. Lau. J. [42] explored several applications of profit function through an alternate derivation to characterize production and profit function. According to the study, the firm maximizes profit with the optimized quantities of the variable inputs where maximization of the profit is similar to the normalized profit function. Chen. Chung-Ho [43] et al. presented the supply chain system's optimal profit-sharing model between producers and consumers. According to the study, the order quantity contributes to the average consumer demand and significantly impacts the predicted profits of the supplier and the buyer.

[Updated] [Section: 4.2 Result Analysis]

To establish and explore new business opportunities in today's world, it has become essential to mix technical innovation and knowledge with business. In light of this, the goal of this article was to provide a novel technology element that would enable an organization to increase its profitability. In order to find the consumer group that would guarantee and provide a larger profit to a business, a new feature V has been introduced to the existing LRFM model. It can be noticed clearly that the profit-volume relationship is proportionate, as shown by the preceding figures (figure 11-13), and any business can generate a sizable profit by analyzing the volume, which is a scaled-down representation of the quantity of things purchased by a potential consumer. By emphasizing the amount of product those clients have purchased over a predetermined time frame or a predetermined number of visits, it helps any business identify a group of valuable customers.

Reviewer#3, Concern # 2: Moreover, the authors provide Fig 7 to represent the relationship between each parameter in the model. Authors assert that the newly introduced parameter volume (V) has a very weak relationship with each of the other parameters and is quite strong with itself. Thus, they conclude that V is a unique feature and can not be replaced by other features. However, when we observe that figure, it is clearly seen that the new proposed feature volume (V) directly correlates with both frequency (around 0.8) and monetary (around 0.6). These values are quite high for the correlation analysis. When this is taken into account, it is seen that the new feature does not add any extra dimension to the current model.

Author response: We want to express our sincerest appreciation to the respected reviewer for raising the interesting fact that the newly proposed feature volume (V) directly correlates with both frequency (around 0.8) and monetary (around 0.6). In the customer segmentation problem, it is necessary to find all the relevant factors to group similar customers together. In contrast, in the feature selection problem, we need to eliminate features with high correlation to remove the redundant features. Therefore, even in the existing LRF and LRFM model, the factor length (L) and frequency (F) exist together although these two factors have a higher correlation (around 0.8) [Please see Figure. 7]. The correlation between frequency (F) and monetary (M) is also high (around 0.5).

We proposed to add volume as a new component to the LRFM model with the goal of presenting a comprehensive and appropriate solution for evaluating various datasets. No feature can be said to be indispensable at first, but based on its application to our real-world dataset and the findings, we can say that it will undoubtedly play a key role in conjunction with the current LRFM model. However, we exercised caution while choosing characteristics since if a feature is no longer relevant, the results would be unclear, inaccurate, and noisy, which may have an impact on decision-making. Our goal of advancing business analysis and profit maximization will be successful if a new potential feature can be included because it will enable us to evaluate and make judgments based on more factors that were previously overlooked. By incorporating "Volume" into the current business models, not only were the sectors with potential customers recognized, but also the segments that drove down profits, as shown in Fig. 13. Additionally, the new LRFMV model's segment numbers are higher than those of earlier models, enabling us to easily distinguish various consumer segment types that have profitable business solutions, as shown in Table 7. After considering these effects of volume, it can be considered as a potential feature in spite of having a correlation with frequency and monetary. 

Author action: Updated sections have been marked yellow using the highlighter tool in the revised

Manuscript.

 Following are the changes made to the manuscript in this regard:

[Updated] [Section: 3.5 Correlation of volume with other features]

As an upgraded version of the RFM model, the Length, Recency, Frequency, and Monetary model (also known as the LRFM model) was developed to identify more precise and pertinent consumer groups for profit maximization. The properties of the current LRFM model, as seen in Figure 7, likewise demonstrate a good correlation between length and frequency (0.8) and between monetary and frequency (0.6). Even yet, many researchers utilize this strategy frequently to successfully analyze client behavior across a range of application fields and in spite of having a good correlation with frequency, length has been introduced. So we believe that volume can help to further assess how much revenue boost and marketing strategy can be developed for the superstore industry and contribute to both technical sectors and the business world.

Reviewer#3, Concern # 3: Authors state that the volume highlights the customers who provide more profit to the organization. However, customer profitability is not related to the amount purchased. Monetary (M) and frequency (F) variables explain customer profitability in the standard model. Let's try to explain with an example that the amount purchased does not add an extra dimension and is a meaningless variable in terms of customer segmentation. There is a customer who buys a lot of products, but that customer may only buy certain products from the relevant supermarket that are quite cheap. In other words, the total amount paid is quite less. In this case, when we consider the amount received, how can we call this customer profitable? Or let's think the opposite, let's say that another customer who buys very few products, and the products he buys are quite expensive. In other words, the total amount paid is quite high. In this case, when we consider the amount received, how can we call this customer non-profitable? Therefore, purchased quantity as seen in the example, does not offer an extra dimension in explaining customer behaviour.

Author response: Superstores carry a wide variety of goods and some have a history of being purchased by the same client multiple times on the same day in various quantities. The relationship between profit and purchased quantity and how they can support an efficient consumer behavioral analysis were not looked into and assessed in the RFM and LRFM models. As superstore customers were our target group, it can be said that superstores don't stock particularly expensive items, and most customers choose to buy necessities from superstores rather than luxury goods. There are some expensive items but the customers do not buy them frequently which is the same for the cheaper items. According to the equation of volume, the number of days while transacting in the customer’s life cycle is also measured. Again, the price of the common items frequently bought by the customers does not vary much in the case of the superstores which is different for the electronic shops. We have also included references for this which includes the most common items bought in a superstore and these are mostly items needed for daily groceries. In light of this, we suggested a feature that will enhance their business analysis and, indirectly, assist them in determining their best-selling products for boosting sales. To do so, we relied on our real-world dataset instead of case studies.

Author action: Updated sections have been marked yellow using the highlighter tool in the revised

Manuscript. 

The following changes have been made to the manuscript in this regard:

[Updated] [Section: 4.1 Comparative Analysis]

From comparative analysis, we can see that the LRFMV model is providing a reasonable output compared to the RFM and LRFM models and is ensuring the polarity, which is not present in RFM or LRFM models for any of the three clustering algorithms. We suggested volume, which was determined by assessing the typical number of items a regular superstore customer purchases in a single day. The previously discovered average quantity was divided by the entire number of days in the customer's brief period of visitation in order to obtain the final average as the volume in a given time frame. There are some expensive products in the superstores as well as very cheap products. However, the price of the most common items frequently bought by the customers are usually daily groceries and do not vary much3,4,5. Additionally, the LRFMV model has one additional cluster while using the K-Medoids algorithm, to increase profitability and the overall profit margin is higher than other models, where both RFM and LRFM models do not have the additional cluster, which also guarantees superior clustering quality. In the bar chart of figure 9, which compares the profitability of three distinct models, it can be seen that the LRFMV model's cluster which is represented by the green bars makes the most profits for clusters 0, 1, and 2 compared to LRFM and RFM models for Mini batch K-Means Algorithm. Furthermore, while the LRFMV model uses K-Means Algorithm to produce a structured output, RFM and LRFM models reveal conflicting profit analyses for various clusters. We can assert that this has occurred in order to add volume to the existing models because the LRFMV model is producing a passable output when compared to the RFM and LRFMV models.

---

## [Editor Report · Decision Letter 2]

5 Dec 2022

LRFMV: An Efficient Customer Segmentation Model for Superstores

PONE-D-21-32764R2

Dear Dr. Md. Golam Rabiul Alam,

We’re pleased to inform you that your manuscript has been judged scientifically suitable for publication and will be formally accepted for publication once it meets all outstanding technical requirements.

Kind regards,

Vincenzo Basile, PhD

Academic Editor

PLOS ONE